# Deep tissue space-gated microscopy via acousto-optic interaction

Mooseok Jang [1,2,3,5]*, Hakseok Ko[1,2,5], Jin Hee Hong[1,2], Won Kyu Lee[4], Jae-Seung Lee [4] & Wonshik Choi [1,2]*

To extend the imaging depth of high-resolution optical microscopy, various gating operations —confocal, coherence, and polarization gating—have been devised to filter out the multiply scattered wave. However, the imaging depth is still limited by the multiply scattered wave that bypasses the existing gating operations. Here, we present a space gating method, whose mechanism is independent of the existing methods and yet effective enough to complement them. Specifically, we reconstruct an image only using the ballistic wave that is acousto-optically modulated at the object plane. The space gating suppresses the multiply scattered wave by 10–100 times in a highly scattering medium, and thus enables visualization of the skeletal muscle fibers in whole-body zebrafish at 30 days post fertilization. The space gating will be an important addition to optical-resolution microscopy for achieving the ultimate imaging depth set by the detection limit of ballistic wave.

[1] Center for Molecular Spectroscopy and Dynamics, Institute for Basic Science (IBS), 145 Anam-ro, Seongbuk-gu, Seoul 02841, Korea. [2] Department of Physics, Korea University, 145 Anam-ro, Seongbuk-gu, Seoul 02841, Korea. [3] Department of Bio and Brain Engineering, KAIST, 291 Daehak-ro, Yuseong-gu, Daejeon 34141, Korea. [4] Department of Materials Science and Engineering, Korea University, 145 Anam-ro, Seongbuk-gu, Seoul 02841, Korea. [5] These authors contributed equally: Mooseok Jang, Hakseok Ko *email: mooseok@kaist.ac.kr; wonshik@korea.ac.kr

Improving the imaging depth of high-resolution optical microscopy has been a long-standing goal in the field of bioimaging due to its potential impact on biological studies and optical diagnostics[1]. For ideal diffraction-limited imaging, the main strategy is to detect the so-called ballistic wave that propagates straight through a scattering medium and carries intact object information. However, this ballistic wave is quickly obscured by multiply scattered waves even at a shallow depth as its intensity decays exponentially with distance traveled in a scattering medium due to multiple light scattering. To extend the imaging depth, the prevailing approach so far has been to filter out the multiply scattered waves by applying various gating operations, such as confocal[2,3], time (or coherence)[4–7], and polarization gating[8,9]. For example, optical coherence tomography, one of the most successful biomedical imaging modalities, greatly extends imaging depth by combining all these gating operations[7,10,11]. Similarly, spatial correlation within a time-gated transmission or reflection matrix has recently been used to selectively extract image information[12,13]. Furthermore, various adaptive optics approaches have been proposed to maintain the effectiveness of gating operations in spite of sample-induced aberration[14–17].

Even with these substantial advances, the imaging depth of high-resolution optical microscopy has not yet reached the detection limit set by the dynamic range of state-of-the-art sensor technology. The ballistic wave is, in principle, detectable even at depths >15 $l_s$ in an epi-detection geometry (where $l_s$ is the scattering mean free path of the scattering medium) if an image sensor of high dynamic range (e.g., 1:10$^4$) is used in conjunction with interferometric detection converting an intensity recording into an electric field measurement[10,12,13,18,19]. Currently, the imaging depth limit is instead set by the competition between the ballistic wave and the multiply scattered wave that bypasses the existing gating operations. The residual multiply scattered wave can be significantly stronger than the ballistic wave well before reaching the detection limit[10,19–21]. For instance, the chance that a multiply scattered wave has a similar flight time to a ballistic wave and passes through a time gating of finite width increases with imaging depth. Likewise, a large fraction of a multiply scattered wave can pass through a confocal pinhole under conditions of extreme turbidity, and thereby be mistakenly considered as a ballistic wave. In fact, these imperfections of the existing gating methods are partly due to their action being at a detection plane, which is located outside the scattering medium. To reach the detection limit, it is critical to develop an additional gating method whose mechanism is independent of the existing methods and yet effective enough to complement them.

Here, we propose a new gating scheme called space gating. Based on the interferometric detection scheme of previous acousto-optic imaging techniques[22–28], we implement the space gating by selectively measuring the ballistic wave that is modulated by a high-frequency ultrasound focus as small as ~30 μm × 70 μm in size. Unlike confocal or time gating, space gating is directly applied at the object plane inside the scattering medium to reject the multiply scattered wave whose optical path spreads beyond the extent of the ultrasound focus. Therefore, it can remove the multiply scattered wave, which cannot be filtered out by the existing gating operations. Integrating the space gating into the coherent confocal microscopy, we demonstrate imaging of amplitude objects through scattering layers thicker than 23$l_s$ with the optical diffraction-limited resolution of 1.5 μm. Furthermore, by combining the noise rejection capability of space gating with the advantage of coherent treatment of the ballistic wave, we demonstrate the quantitative phase imaging of biological cells fully embedded within a scattering medium. Lastly, we examine the effectiveness of space gating in imaging skeletal muscle structures

of an unstained zebrafish across its entire body. The proposed concept of space gating is an independent and complementary addition to the existing gating operations. It represents an important step toward reaching the fundamental depth limit of diffraction-limited imaging relying on ballistic waves, and opens new possibilities for label-free imaging of biological cells through scattering tissues.

## Results

**Principles.** The concept of space gating combined with confocal gating is illustrated in Fig. 1a. To implement the confocal gating, we illuminated an object plane with a focused laser beam and detected the transmitted field at the position $\mathbf{r}_d$ conjugate to the illumination point $\mathbf{r}_i$. As shown in Fig. 1a, $\mathbf{r}_i$ and $\mathbf{r}_d$ are the illumination and detection points defined on the planes conjugate to the object plane. We measured the signal only at the sensor pixels (marked with a blue square) conjugate to the focused illumination. This ensures that only the ballistic wave (indicated as green lines in Fig. 1a), which carries the optical diffraction-limited image, contributes to the measured field in the absence of scattering. This scheme is equivalent to the conventional confocal gating, where a physical pinhole is used.

In the presence of scattering, the transmitted field $E(\mathbf{r}_d; \mathbf{r}_i)$ measured at the detection plane is composed of two components: ballistic signal field $E_S(\mathbf{r}_d; \mathbf{r}_i)$ and multiply scattered noise field $E_M(\mathbf{r}_d; \mathbf{r}_i)$ (i.e., $E(\mathbf{r}_d; \mathbf{r}_i) = E_S(\mathbf{r}_d; \mathbf{r}_i) + E_M(\mathbf{r}_d; \mathbf{r}_i)$). In deep tissue imaging, the multiply scattered wave often obscures the ballistic wave and limits the imaging depth of diffraction-limited imaging. The space gating aims to selectively suppress the multiply scattered wave based on the fact that it is spatially spread over the wide extent on the object plane (as indicated as blue lines in Fig. 1a), in contrast to the ballistic wave which is tightly confined at the confocal point. The space gating is implemented by setting a spatial window $R_{SG}$ (indicated by the red spot in Fig. 1a) around the confocal point on the object plane in such a way that only the wave transmitted through the gating window contributes to the detected field. This operation selectively rejects the multiply scattered wave traveling outside the spatial window (indicated as the blue dotted lines in Fig. 1a), while leaving the ballistic wave unaffected.

The effect of the space gating can be quantitatively understood by the transfer functions $T_i(\mathbf{r}_o; \mathbf{r}_i)$ and $T_d(\mathbf{r}_o; \mathbf{r}_d)$ describing the optical propagation through the illumination and detection parts of the scattering medium, respectively:

$$T_i(\mathbf{r}_o; \mathbf{r}_i) = S(\mathbf{r}_o; \mathbf{r}_i)e^{-\frac{L_i}{2l_s}} + M_i(\mathbf{r}_o; \mathbf{r}_i) \qquad (1)$$

$$T_d(\mathbf{r}_o; \mathbf{r}_d) = S(\mathbf{r}_o; \mathbf{r}_d)e^{-\frac{L_d}{2l_s}} + M_d(\mathbf{r}_o; \mathbf{r}_d). \qquad (2)$$

The subscripts i and d indicate the illumination and detection parts of the sample, as indicated in Fig. 1a. $T_i(\mathbf{r}_o; \mathbf{r}_i)$ is the complex field amplitude at $\mathbf{r}_o$ on the object plane for the illumination of a unity amplitude field originated from $\mathbf{r}_i$, and $T_d(\mathbf{r}_o; \mathbf{r}_d)$ is defined likewise for the detection part. $L$ and $l_s$ are the thickness and scattering mean free path of the sample, respectively, and $S(\mathbf{r}_o; \mathbf{r}_i)$ and $S(\mathbf{r}_o; \mathbf{r}_d)$ denote the transfer functions of ballistic waves, which are the intrinsic point spread functions (PSFs) of the optical system (Fig. 1b). For simplicity, we assume unity magnification from the planes of $\mathbf{r}_i$ and $\mathbf{r}_d$ to the object plane. $M_i(\mathbf{r}_o; \mathbf{r}_i)$ and $M_d(\mathbf{r}_o; \mathbf{r}_d)$ denote the transfer functions of multiply scattered waves, which extend over a wide area on the object plane (Fig. 1c).

The transmitted field $E(\mathbf{r}_d; \mathbf{r}_i)$ without the space gating can be described with the two transfer functions:

$$E(\mathbf{r}_d; \mathbf{r}_i) = \int_R T_i(\mathbf{r}_o; \mathbf{r}_i)T_d(\mathbf{r}_o; \mathbf{r}_d)d\mathbf{r}_o, \qquad (3)$$

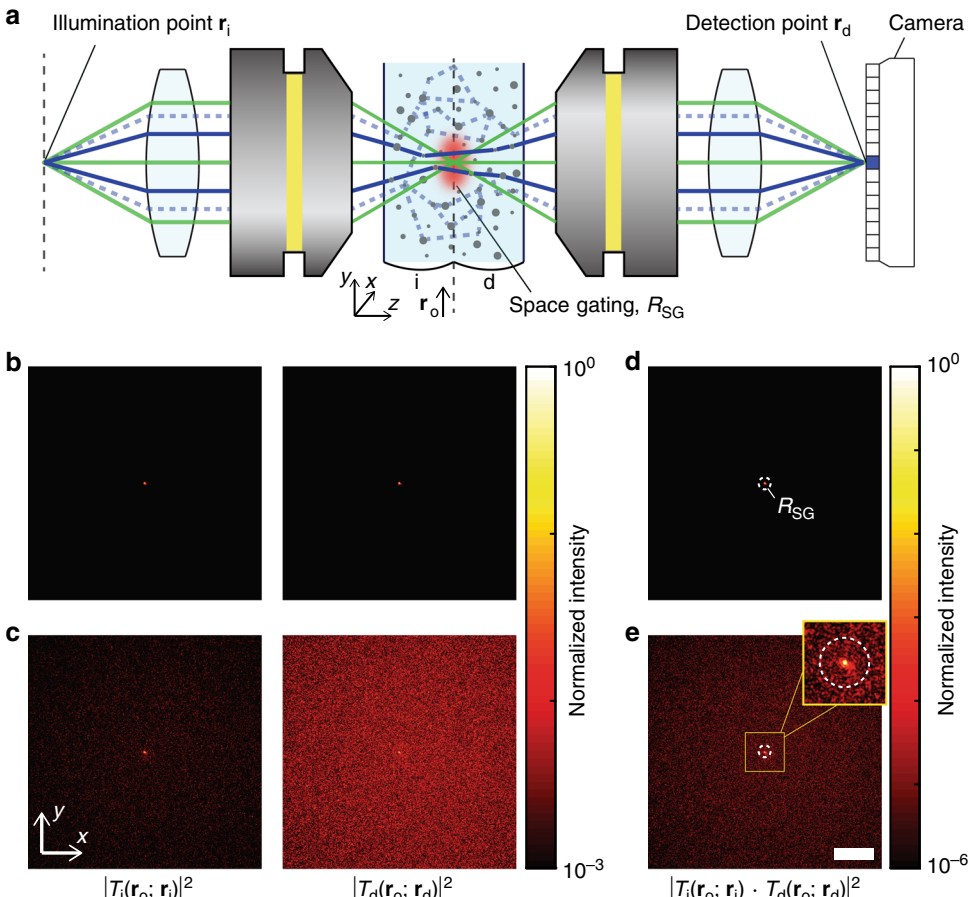

**Fig. 1 Principle of space gating. a** Schematic of the imaging principle. Conventional confocal imaging method relies on the ballistic waves shown as green lines. When optical inhomogeneity is introduced, the intensity of the ballistic wave exponentially decreases with depth, and the multiply scattered wave (shown as solid blue and dotted blue lines) may obscure the ballistic wave. By implementing space gating at the object plane using an acousto-optic effect (indicated as a red spot), the multiply scattered wave that travels outside the acoustic focus (dotted blue lines) can be rejected, which in turn improves the ratio of the ballistic wave to the multiply scattered wave at the sensor element (marked as a blue pixel), whose position is conjugate to the illumination point $\mathbf{r}_d \sim \mathbf{r}_i$. **b** Intensity maps of illumination and detection transfer functions in a confocal gating scheme (where $\mathbf{r}_d \sim \mathbf{r}_i$), with respect to $\mathbf{r}_o$ on the object plane for a transparent medium. **c** Same as **b**, but in the presence of scattering. The optical thicknesses on the illumination and detection sides were $L_i/l_s = 10.6$ and $L_d/l_s = 12.8$, respectively. **d** Contribution map, $|T_i(\mathbf{r}_o; \mathbf{r}_i)T_d(\mathbf{r}_o; \mathbf{r}_d)|^2$, with respect to $\mathbf{r}_o$ calculated from the transfer functions in **b**. **e** Same as **d**, but in the presence of scattering, calculated from **c**. Scale bar: 100 μm.

where $R$ covers the entire object plane. This equation is subject to the assumption that the multiple scattering between the illumination and detection parts of the scattering medium is negligible, which is largely the case for the highly anisotropic scattering medium. Note that the multiple scattering within the illumination and detection parts of the scattering medium is already accounted for in $T_i(\mathbf{r}_o; \mathbf{r}_i)$ and $T_d(\mathbf{r}_o; \mathbf{r}_d)$. By inserting Eqs. (1) and (2) into Eq. (3), we obtain the signal field and the noise field as follows:

$$E_S(\mathbf{r}_d; \mathbf{r}_i) = \int_R S(\mathbf{r}_o; \mathbf{r}_i) S(\mathbf{r}_o; \mathbf{r}_d) e^{-\frac{(L_i+L_d)}{2l_s}} d\mathbf{r}_o \qquad (4)$$

$$E_M(\mathbf{r}_d; \mathbf{r}_i) = \int_R \left[ S(\mathbf{r}_o; \mathbf{r}_i) e^{-\frac{L_i}{2l_s}} M_d(\mathbf{r}_o; \mathbf{r}_d) + S(\mathbf{r}_o; \mathbf{r}_d) e^{-\frac{L_d}{2l_s}} M_i(\mathbf{r}_o; \mathbf{r}_i) \right.$$
$$\left. + M_i(\mathbf{r}_o; \mathbf{r}_i) M_d(\mathbf{r}_o; \mathbf{r}_d) \right] d\mathbf{r}_o.$$
$$(5)$$

See Supplementary Note 1 for the discussion on the relative magnitude among the signal field and the three terms in the noise field. The multiplication of $S(\mathbf{r}_o; \mathbf{r}_i)$ and $S(\mathbf{r}_o; \mathbf{r}_d)$ in the signal field

of Eq. (4) describes the confocal action. The multiplication of the two transfer functions $T_i(\mathbf{r}_o; \mathbf{r}_i)T_d(\mathbf{r}_o; \mathbf{r}_d)$, shown in Fig. 1d, e, describes how much each point $\mathbf{r}_o$ on the object plane contributes to the light propagation from the illumination point $\mathbf{r}_i$ to the detection point $\mathbf{r}_d$. Mathematically, the space gating sets the spatial window $R_{SG}$ around the confocal point, as indicated by the white dotted lines in Fig. 1d, e. Therefore, the measured field with the space gating $E^{SG}(\mathbf{r}_d; \mathbf{r}_i) = E_S^{SG}(\mathbf{r}_d; \mathbf{r}_i) + E_M^{SG}(\mathbf{r}_d; \mathbf{r}_i)$ can be derived by Eqs. (3), (4), and (5) after reducing the integration range from $R$ to $R_{SG}$. Because the signal field of Eq. (4) and the first two terms in the noise field of Eq. (5) involve the ballistic propagation confined to the confocal point, only the $M_i(\mathbf{r}_o; \mathbf{r}_i)M_d(\mathbf{r}_o; \mathbf{r}_d)$ term in the noise field is reduced by the space gating. Considering that this term is dominant in determining the noise field (see Supplementary Note 1 for further explanation), the noise suppression factor $\eta$ can be estimated as:

$$\eta = |E_M(\mathbf{r}_d; \mathbf{r}_i)|^2 / \left| E_M^{SG}(\mathbf{r}_d; \mathbf{r}_i) \right|^2 \sim \min\left(w_{M_i}, w_{M_d}\right)^2 / w_{SG}^2. \quad (6)$$

Here, $w_{Mi}$ and $w_{Md}$ are the effective widths of $|M_i(\mathbf{r}_o; \mathbf{r}_i)|^2$ and $|M_d(\mathbf{r}_o; \mathbf{r}_d)|^2$, respectively. $w_{SG}$ is the width of $R_{SG}$ set by the acoustic focus size in our experiment. For biological tissues, $w_{Mi}$

and $w_{Md}$ typically range from hundreds of microns to millimeters when $L/l_s \sim 10$ (see Supplementary Fig. 1 for the detailed analysis on $w_{Mi}$ and $w_{Md}$). Therefore, we can expect $\eta > 100$ if the size of the space gating $w_{SG}$ is as small as tens of microns, as is the case with a high-frequency acoustic focus.

To see the effect of space gating on the imaging depth of the optical diffraction-limited imaging, we define the imaging fidelity by the contrast of ballistic wave: $\tau = |E_S(\mathbf{r}_d; \mathbf{r}_i)|^2/|E_M(\mathbf{r}_d; \mathbf{r}_i)|^2$ without space gating, and $\tau^{SG} = \left|E_S^{SG}(\mathbf{r}_d; \mathbf{r}_i)\right|^2/\left|E_M^{SG}(\mathbf{r}_d; \mathbf{r}_i)\right|^2$ with space gating. When the imaging fidelity is sufficiently $>1$, the ideal optical diffraction-limited imaging is achieved as the detected field is mostly comprised of the ballistic wave. When increasing the imaging depth, the spatial resolution remains close to the optical diffraction limit of the confocal imaging system, while the contrast of the ballistic wave is reduced due to the exponential decay of the ballistic wave. The space gating improves the imaging fidelity by a factor of $\eta$, i.e., $\tau_{SG} = \eta \times \tau$. Considering the exponential decay of the intensity of ballistic wave, the imaging depth increases logarithmically with $\eta$. More specifically, the noise suppression effect can compensate the additional decay of ballistic wave by the increased imaging depth, i.e., $\eta \times e^{-\Delta L/l_s} = 1$, where $\Delta L$ is the gain in the imaging depth by the space gating. Therefore, $\eta$ is translated into $\Delta L = l_s \times \log \eta$. For $\eta > 100$, we can expect the gain in imaging depth $\Delta L$ of $>5l_s$. We provide further analysis on the relation of the imaging depth to the size of the acoustic focus and the optical wavelength in the Supplementary Note 2.

**Confocal imaging setup with acousto-optic space gating.** Figure 2a shows the experimental configuration of the confocal imaging system integrated with a high-frequency acousto-optic space gating (see Methods for details of the setup). Our scheme of space gating is based on an interferometric detection method similar to the previously demonstrated ultrasound-modulated optical tomography[24,25,27–30]. When light propagates through the oscillating pressure field at the acoustic focus, a fraction of the light is modulated by the frequency of $f_{US}$. Then, the reference beam with the frequency of $f_0 + f_{US}$ form a static interference pattern with the modulated wave. The complex field of the modulated wave is then selectively measured using four-step phase-shifting interferometry[31]. See Supplementary

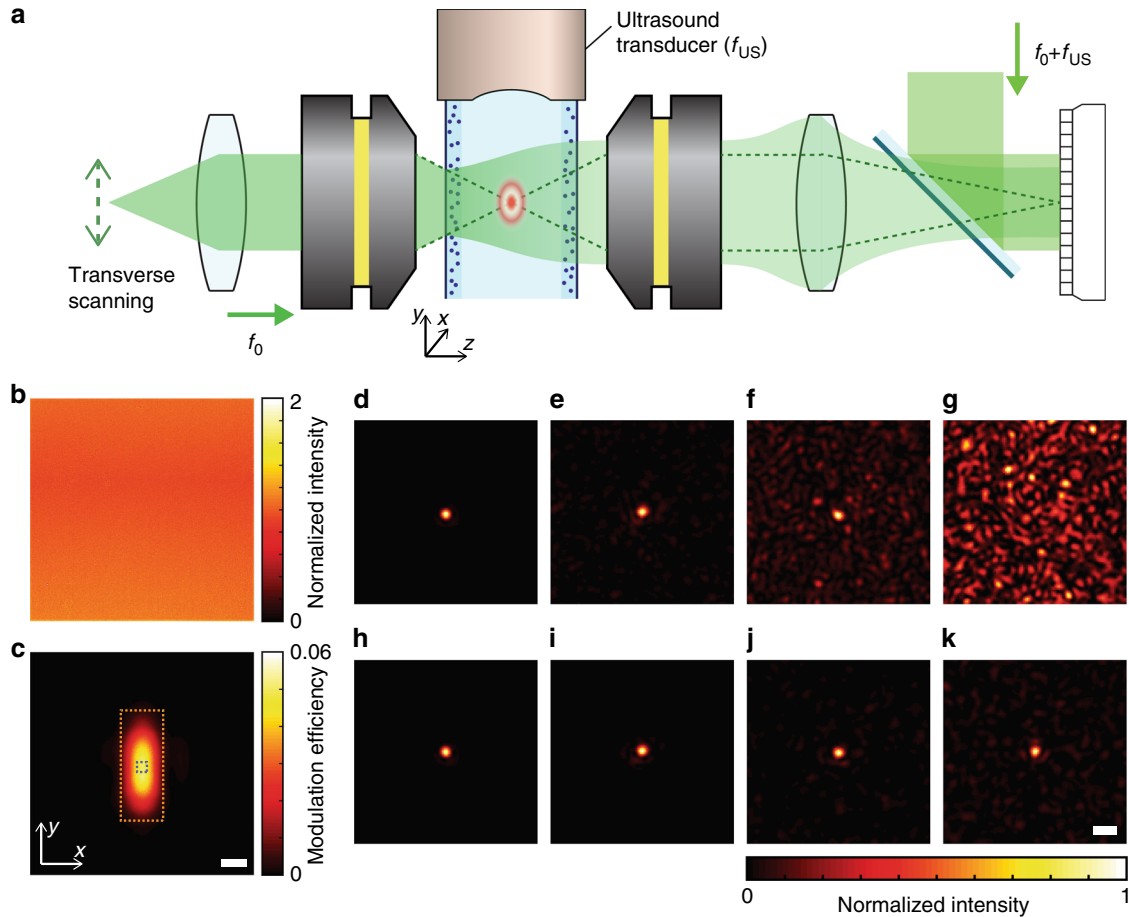

**Fig. 2 Confocal imaging setup with acousto-optic space gating. a** Focused acoustic beam modulates the frequency of the incident focused illumination beam, whose optical frequency is $f_0$. Only the frequency-modulated wave through the region of the space gating (i.e., acoustic focus) is measured at the sensor plane by using a phase-shifting interferometry, where the frequency of the reference beam is set to that of the acoustically modulated optical wave $f_0 + f_{US}$. **b** Average intensity map for 900 planar illuminations at different incidence angles through a transparent medium without space gating. The entire object plane contributes to the detected signal. The intensity map is normalized to the mean intensity. **c** Same as **b**, but with space gating. With the space gating, only the region inside the gating window (i.e., the acoustic focus) contributes to the detected signal. The intensity map was normalized such that it represents the acoustic modulation efficiency. Scale bar: 30 μm. **d–g** Point spread functions (PSFs) $|E(\mathbf{r}_d; \mathbf{r}_i)|^2$ measured on the detection plane without space gating, when the optical thicknesses of the input and output layers were (0, 0), (6.9, 10.6), (6.9, 12.8), and (10.6, 12.8), respectively. **h–k** PSFs $|E_{SG}(\mathbf{r}_d; \mathbf{r}_i)|^2$ with space gating for the corresponding scattering layers to **d–g**. PSFs were normalized to their maximum intensities. Scale bar: 5 μm.

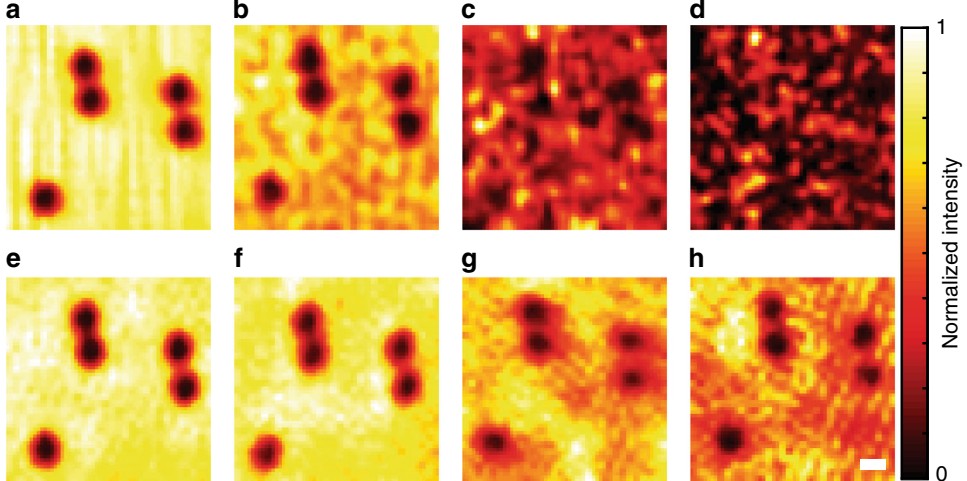

**Fig. 3 Demonstration of space gating in confocal imaging.** Images were reconstructed by scanning 900 points within a 16.1 × 16.1 μm² field of view. **a–d** Reconstructed intensity images of 2-μm gold-coated microspheres without space gating when the optical thicknesses of the input and output layers were (0, 0), (6.9, 10.6), (6.9, 12.8), and (10.6, 12.8), respectively. **e–h** Reconstructed images with space gating for the same configurations as **a–d**. Images were normalized to their maximum intensities. Scale bar: 2 μm.

Figs. 2 and 3 for the detailed experimental setup, and the electrical signal flow for the acousto-optic measurements, respectively.

We first confirmed the spatial extent of the acousto-optic space gating (see Methods for measurement details). Without the space gating, the intensity map was uniform across the field of view (Fig. 2b). When the space gating was applied, only the wave traveling through the acousto-optic gating window, $R_{SG}$, was visible (Fig. 2c). The full-width at half maximum (FWHM) of $R_{SG}$ were 29 and 72 μm along the $x$- and $y$-axes, respectively. Because the acoustic impedance of fat, water, blood, and muscle does not differ >10%, soft tissues may be considered homogenous for the acoustic wave. This guarantees the diffraction-limited confinement of acoustic focus. In the case of hard tissues, such as bone and teeth, they block the propagation of acoustic wave as their acoustic impedances are five times as large as that of soft tissues. This is a common limitation for all the ultrasound-based applications in biology and medicine. The gating contrast, measured by the ratio between the average intensity inside the blue box and outside the orange box in Fig. 2c, was ~100, and the modulation efficiency around the focal area (see Methods for calculation details) was 5%, which has been optimized through the precise synchronization between the laser pulses and acoustic pulses. Note that this acousto-optic modulation efficiency does not affect the signal to noise ratio $\tau^{SG}$ or the noise suppression factor $\eta$ because both the signal and noise are subject to the same modulation efficiency.

Figure 2d–k presents the PSFs $|E(\mathbf{r}_d; \mathbf{r}_i)|^2$ and $|E^{SG}(\mathbf{r}_d; \mathbf{r}_i)|^2$ without and with space gating, respectively, measured at the detector plane. Figure 2d, h is the intrinsic system PSFs through a transparent medium composed of polyacrylamide (PAA) gel. The FWHM of the foci, which dictate the imaging resolution, were measured to be 1.5 μm with or without space gating. In Fig. 2e–g, i–k, we introduced an optical inhomogeneity using scattering poly (dimethylsiloxane) (PDMS) layers on the input and output surfaces of a sample cuvette (see Methods section for details of sample preparation). The distance between the input/output surfaces to the object plane was ~4 mm. The optical thicknesses of the input and output layers ($L_i/l_s$, $L_d/l_s$) were respectively (6.9, 10.6), (6.9, 12.8), and (10.6, 12.8) for each of the Fig. 2e–g, i–k. The ballistic wave appeared as a peak at the detection point $\mathbf{r}_d$ conjugate to the illumination point $\mathbf{r}_i$, and the fluctuating

background of the multiply scattered wave spread across the detector plane. The imaging fidelity $\tau$ and $\tau_{SG}$ were experimentally determined as the averaged intensity ratio of the peak to the fluctuating background (see Methods section for details). For instance, for the case of ($L_i/l_s$, $L_d/l_s$) ~ (10.6, 12.8), $\tau$ was ~0.1 while $\tau_{SG}$ was ~30, from which we can expect only space-gated imaging to properly provide a diffraction-limited resolution in this scattering regime.

**Amplitude imaging through a scattering medium**. We first performed space-gated imaging of amplitude objects through scattering layers of various thicknesses (Fig. 3; see Methods section for details of the objects and the imaging procedure). The reconstructed images without and with space gating are shown in Fig. 3a–h, respectively. The optical thicknesses of the scattering layers ($L_i/l_s$, $L_d/l_s$) were (0, 0), (6.9, 10.6), (6.9, 12.8), and (10.6, 12.8) (the same configuration as for the PSF measurements in Fig. 2d–k). In a relatively weak scattering regime (Fig. 3a, b, e, f), both methods yielded images of the amplitude objects, although there was considerable background fluctuation in the conventional confocal imaging. When scattering became stronger (Fig. 3c, d, g, h), only the space-gated confocal imaging provided resolved images of the amplitude objects. It is remarkable that, with the aid of space gating, the objects could be clearly resolved even in the highly scattering regime of $(L_i + L_d)/l_s > 23$, while the conventional method presented only randomly fluctuating noise dominated by the multiply scattered wave. The imaging results are in good agreement with the PSFs measured in Fig. 2d–k, in the sense that well-resolved images were reconstructed only when the spot contrast ($\tau$ and $\tau_{SG}$) was sufficiently larger than unity. Interestingly, for the case of ($L_i/l_s$, $L_d/l_s$) ~ (6.9, 12.8), the reconstructed image (Fig. 3c) was significantly degraded even though the ballistic spot was distinctively visible (i.e., $\tau = 9.1 > 1$ as shown in Fig. 2f). This is because $E_M(\mathbf{r}_d; \mathbf{r}_i)$ of relatively small amplitude can cause a large fluctuation in $|E_S(\mathbf{r}_d; \mathbf{r}_i) + E_M(\mathbf{r}_d; \mathbf{r}_i)|^2$ depending on the relative phase of $E_M(\mathbf{r}_d; \mathbf{r}_i)$ to $E_S(\mathbf{r}_d; \mathbf{r}_i)$ (see Supplementary Fig. 4 for the quantitative analysis about this effect).

**Noise suppression factor achieved by space gating**. To elucidate how the effect of space gating varied depending on the degree of scattering, we estimated $\tau$ and $\tau_{SG}$ over a wide range of total

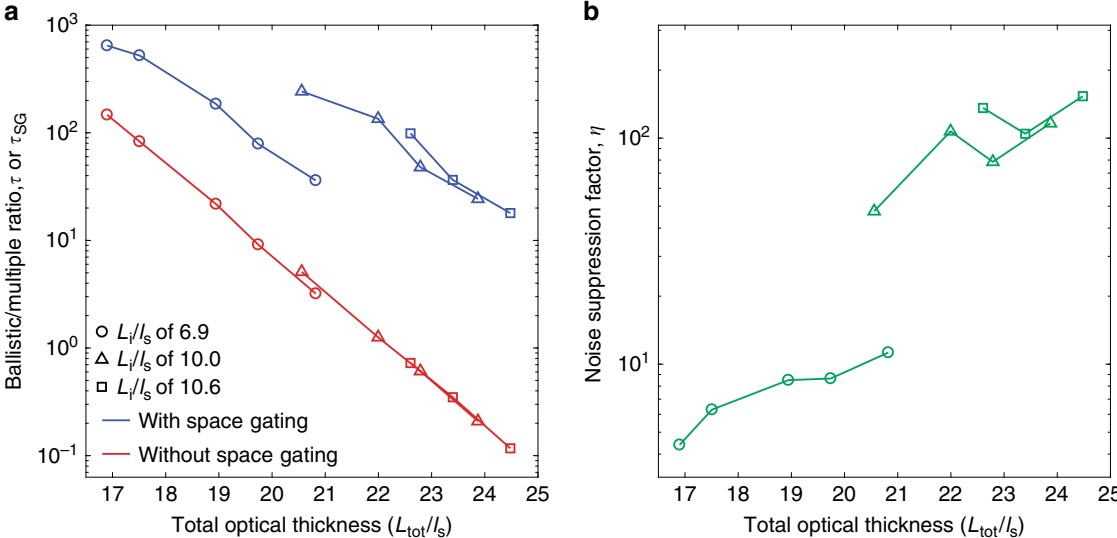

**Fig. 4 Noise suppression efficiency of space gating. a** Ratio of the ballistic wave to the multiply scattered wave with space gating ($\tau_{SG}$, blue) and without space gating ($\tau$, red) as a function of the total optical thickness, $L_{tot}/l_s$. Circular, triangular, and rectangular markers indicate cases of input layer optical thicknesses, $L_i/l_s$, fixed to 6.9, 10.0, and 10.6, respectively. The optical thickness of the output layer was also varied for each case. **b** Noise suppression factor $\eta$ of space gating. $\eta$ was obtained from $\tau_{SG}/\tau$ in **a**.

optical thickness, $L_{tot}/l_s$ with $L_{tot} = L_i + L_d$ (see Supplementary Fig. 5 for the measured PSFs of $|E(\mathbf{r}_d; \mathbf{r}_i)|^2$ and $|E^{SG}(\mathbf{r}_d; \mathbf{r}_i)|^2$ for all combinations of input and output layers). We fixed $L_i/l_s$ to 6.9, 10.0, and 10.6 and varied $L_d/l_s$ for each case of input layer. For convenience of analysis, we set $L_i/l_s < L_d/l_s$ in our experiments, so that $w_{Mi} < w_{Md}$ for all cases. In Fig. 4a, three lines with different markers show $\tau$ and $\tau_{SG}$ for the three cases of $L_i/l_s$. In every case, $\tau_{SG}$ lies well above $\tau$, proving the effectiveness of space gating. $\tau$ monotonically decreased with $L_{tot}/l_s$, and its behavior was generally dictated by the exponential decay of the ballistic wave because the decay of the multiply scattered wave was much slower. On the contrary, $\tau_{SG}$ was highly dependent on $L_i/l_s$. For instance, at $L_{tot}/l_s = 21$, $\tau_{SG}$ was 36 for $L_i/l_s = 6.9$ and 240 for $L_i/l_s = 10.0$. This supports our theoretical prediction in Eq. (6) that the effect of space gating is mainly determined by $w_{Mi}(<w_{Md})$, which was set by $L_i/l_s$, rather than $L_{tot}/l_s$.

Figure 4b presents $\eta$, obtained from $\tau_{SG}/\tau$. $\eta$ ranged from 4.4 to 150 depending on the combination of the input and output scattering layers. As predicted in Eq. (6), $\eta$ was largely determined by $L_i/l_s$, and $L_d/l_s$ had a marginal impact on $\eta$ as $L_i/l_s < L_d/l_s$. At $L_i/l_s = 6.9$, $\eta$ was in the range of 4–11, while it increased to 47–150 when $L_i/l_s$ was increased to 10 or 10.6. Similar to $\tau_{SG}$, even for the same $L_{tot}/l_s$ (e.g., at $L_{tot}/l_s = 21$ in our experiments), $\eta$ can vary significantly, implying that the effect of space gating is highly dependent on the axial position of the object plane within a homogeneously scattering medium. The maximum noise suppression factor we observed was $\eta = 150$ for the configuration of $(L_i/l_s, L_d/l_s) = (10.6, 13.9)$. See Supplementary Fig. 6 for the transfer functions, $T_i(\mathbf{r}_o; \mathbf{r}_i)$ and $T_d(\mathbf{r}_o; \mathbf{r}_d)$, of all scattering layers, and Supplementary Fig. 7 for $\tau$, $\tau_{SG}$, and $\eta$ that were estimated from $T_i(\mathbf{r}_o; \mathbf{r}_i)$ and $T_d(\mathbf{r}_o; \mathbf{r}_d)$ based on the model presented in the Principles section.

**Coherent imaging of objects embedded inside a turbid medium**. Here, instead of having the gap between the scattering layer and the object plane, such as in previous studies[32,33] and our proof-of-concept experiments in Figs. 2 and 3, we considered the fully embedded amplitude objects within a scattering medium

(Fig. 5a) and performed space-gated imaging to verify that our imaging scheme is robust against the small speckle grains inside an acoustic focus. .We confirmed that, at the object plane, the width of speckle grain was 280 nm, which is about half the wavelength (see Methods section for details of sample preparation and determination of speckle size). While the image was completely scrambled by multiple scattering without space gating (Fig. 5b), the object was clearly resolved with space gating (Fig. 5c). $\tau_{SG}$ and $\tau$ were estimated to be 260 and 1.1 from the PSFs (Supplementary Fig. 8), leading to a noise suppression factor $\eta$ of 240.

By leveraging the noise suppression capability of space gating with the coherent treatment of a ballistic wave, we demonstrate the unique capability of space gating—the quantitative phase imaging of human red blood cells completely embedded within a scattering medium (see Methods section for details of sample preparation). As shown in Fig. 5d, only the speckled phase pattern was visible without space gating due to the dominance of the multiply scattered wave over the ballistic wave. In contrast, our method revealed the phase delay associated with the morphology of the red blood cells embedded within the scattering medium (Fig. 5e). To our knowledge, this is the first experimental demonstration of the quantitative phase imaging of biological cells embedded within such a thick scattering medium. This opens a new venue for interrogating transparent biological cells within small animals or organs, with no use of exogenous contrast agents.

**Demonstration of deep imaging within a 30-dpf zebrafish**. To prove the effectiveness of space gating in the context of imaging within intact biological tissues, we performed imaging of whole-body zebrafish at 30 days post fertilization (dpf) and intentionally chose the imaging plane behind the spinal cord to demonstrate the capability of space gating in a more realistic situation, where the complex structures, such as skin, bone, muscle, and organs are heterogeneously distributed between the imaging plane and the imaging objective lens. We note that high-resolution fluorescence imaging for whole-body studies is restricted to young

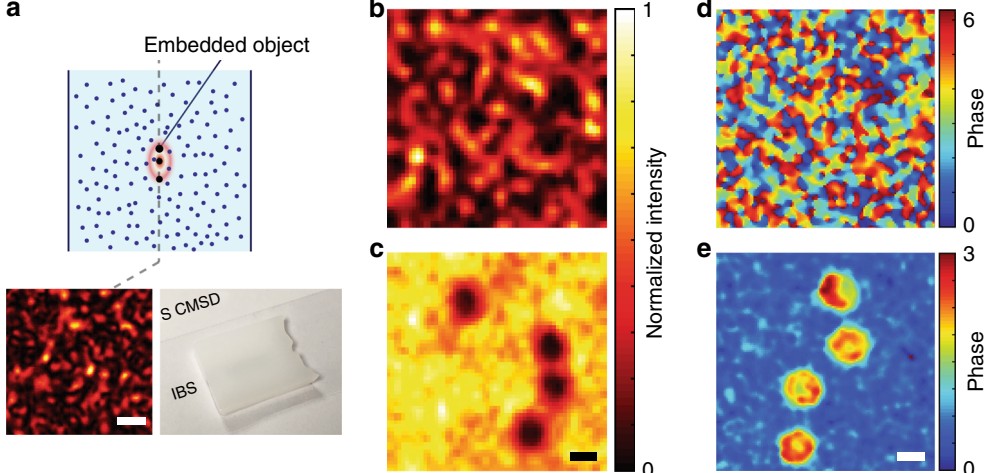

**Fig. 5 Coherent imaging of objects fully embedded within a scattering medium. a** Schematic of the sample configuration. The bottom left inset shows the speckle pattern measured right at the object plane after removing the right hand side of the scattering medium (inset scale bar: 1 μm), and the bottom right inset shows a photograph of the scattering medium. The optical thickness of the scattering slab was 21.0. **b, c** Reconstructed images of 2-μm gold-coated microspheres embedded within the scattering medium without and with space gating, respectively. With the noise suppression factor $\eta = 240$ by the space gating, the gold-coated microspheres were clearly resolved. Images were normalized to their maximum intensities. Scale bar: 2 μm. **d, e** Reconstructed phase images of human red blood cells embedded within the same scattering medium used in **b** and **c** without and with space gating, respectively. With space gating, the size and the morphology of the red blood cells can be obtained from the phase map. Scale bar: 5 μm.

zebrafish of a few days after fertilization due to its shallow imaging depth[34–36]. The 30-dpf zebrafish was ~560-μm thick within the transverse section across the head-trunk region, and the imaging plane was placed 180 μm behind the spinal cord as depicted in Fig. 6a. Therefore, the imaging depth was 460 μm from the surface of zebrafish. In this region, three important structural features of skeletal muscle fibers manifest in a conventional hematoxylin and eosin (H&E) stained histological section: the myosepta that separate and support the blocks of muscle fibers (indicated as the dotted yellow lines in Fig. 6b), the obliquely arranged muscle fibers in between the myosepta (indicated with the dotted white arrow in Fig. 6b), and the alternating light and dark bands (i.e., sarcomere), called I-bands and A-bands, along the muscle fibers (i.e., along the dotted white arrow in Fig. 6b).

We reconstructed the image of skeletal muscle fibers over a large field of view of 780 μm × 200 μm by stitching multiple images (see Methods section for the detailed procedure for the coherent image stitching method). Without space gating, the structural features were not readily visible as multiply scattered waves introduced speckle-like artifacts (Fig. 6c, e–g). The effect of this noise becomes more noticeable toward the anterior side, as the internal structures of zebrafish becomes more complicated within the anterior side of head-trunk junction. In contrast, space-gated imaging provides the distinctive features of myosepta, muscle fibers, and sarcomere (Fig. 6d, h–j). Therefore, with space-gated imaging, one may determine some important structural parameters, such as the position and angle of myosepta, and the sarcomere length (see Supplementary Fig. 9 for one-dimensional profiles)[37]. Additionally, the space-gated image (Fig. 6d) clearly presents the attachment point of muscle fibers and occipital bone, which also appears in the histological image in Fig. 6b (indicated as the dotted red line).

As space gating reduces the random-phase noise of multiply scattered wave, the phase information of muscle fibers becomes clearly visible (Fig. 6k, l). The phase information was particularly useful as a complementary information to determine the discrete muscle fiber from a network of interwoven muscle fibers,

illustrating the benefit of phase imaging within a scattering tissue. Phase imaging also allowed us to enhance the contrast of the individual muscle fibers with the reconstruction of a phase-gradient image based on the application of an asymmetric detection scheme[38] to a complex field image (see Supplementary Fig. 10 for the phase map and the corresponding phase-gradient images). We also imaged sponge-like cartilage structures near the head of the whole-body zebrafish and consistently observed the effect of noise rejection by space gating and the benefit of phase information (Supplementary Fig. 11).

## Discussion

Deep tissue space-gated microscopy, as implemented with the simple addition of an acoustic focus to a conventional microscopy, can be used to improve the imaging depth of a wide range of label-free imaging applications that rely on the intrinsic optical absorption and phase-gradient contrast of the specimen. In this work, we demonstrated wide-field-of-view imaging of a whole-body zebrafish and showed that the space gating rejects a significant portion of multiple scattering noise and reveals the important structural features, such as myosepta and sarcomere, even through the spinal cord located at the center of the body. This example illustrates the potential use[39–42] of space gating to achieve histology-like imaging within a scattering tissue without any incision or staining procedure typically required for histological methods[43,44]. Additionally, the coherent nature of space-gated microscopy enables visualizing biological phase objects within deep tissue, which might directly benefit electrophysiology experiments.

The proposed space gating method is the first acousto-optic imaging approach relying on the selective and coherent detection of the ballistic waves. Therefore, its resolution is dictated by the ideal diffraction limit of the optical system, rather than the diffraction limit of the acoustic system. Although our scheme of space gating shares some components with the conventional ultrasound-modulated optical tomography[24–28], our space-gated microscopy uses the acousto-optic effect in a completely different

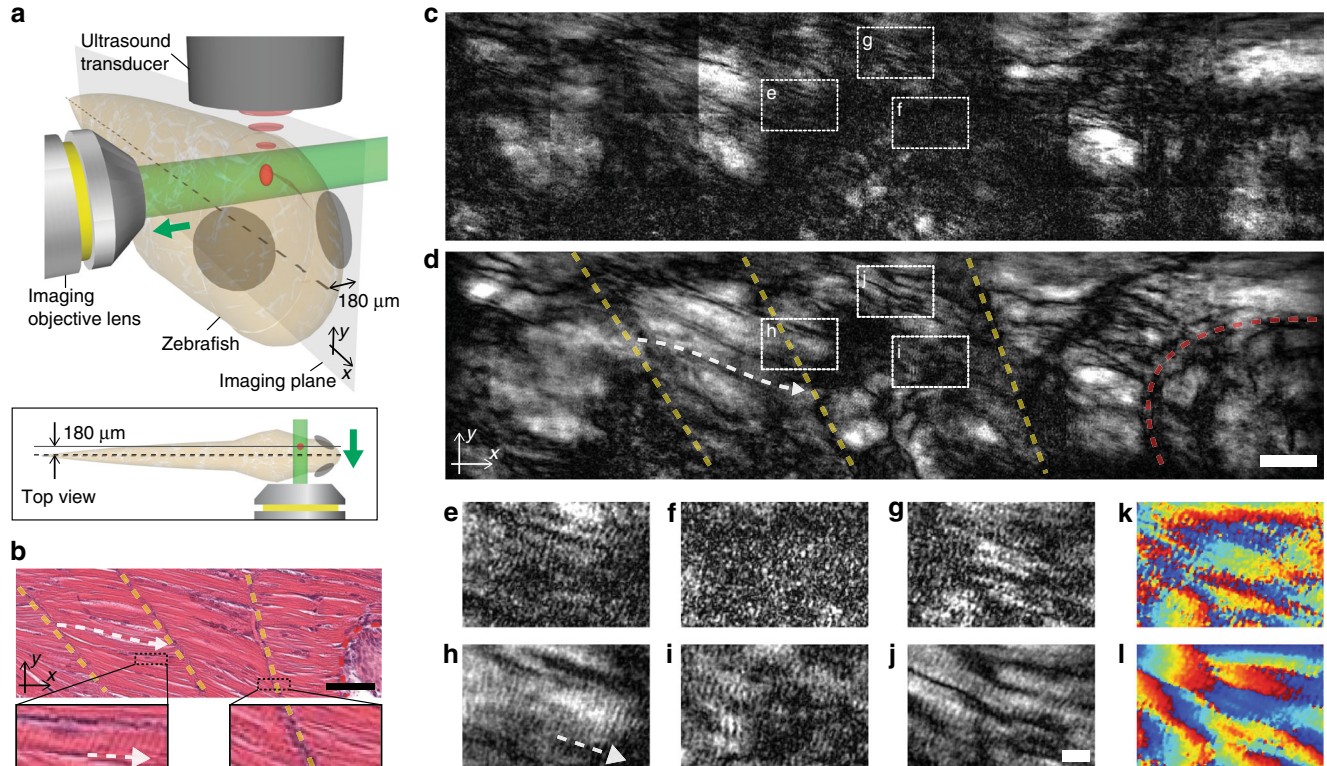

**Fig. 6 Demonstration of space-gated imaging within a 30-dpf zebrafish. a** Schematic of the imaging configuration for a whole-body zebrafish. The skeletal muscle structure of the head-trunk junction was imaged. We chose the imaging plane 180 μm behind the spinal cord to have the complex structures, such as skin, thick muscle layer, spinal cord, and cartilage along the beam path between the imaging plane and the imaging objective lens. The bottom inset shows the top view of the imaging configuration. **b** A typical high-resolution histological section of the skeletal muscle fibers at the head-trunk junction. The position of myosepta and the muscle-bone junction are indicated with dotted yellow and red lines, respectively. The dotted white arrow indicates the direction of an obliquely arranged muscle fiber. The bottom left inset shows the magnified view of an individual muscle fibers. The alternating light and dark bands of sarcomere are barely visible along the direction of the white arrow. The bottom right inset shows the magnified view of myosepta. Scale bar: 50 μm. The histological image is adapted from PennState Bio-Atlas database (http://bio-atlas.psu.edu/; http://bio-atlas.psu.edu/view.php?s=64&atlas=73). **c, d** Wide-field imaging without and with space gating, respectively. With space gating, the structural features of the myosepta (dotted yellow lines) and the muscle-bone junction (dotted red line) can be identified. Similar to **b**, the dotted white arrow indicates the direction of an obliquely arranged muscle fiber. Scale bar: 50 μm. **e–j** Magnified views of the regions indicated in **c** and **d**. Space-gated images reveals the repeating unit of muscle fiber (i.e., alternating light and dark bands) arranged along the individual muscle fibers, whose direction is indicated with the white arrow. From the period of the alternating bands, the sarcomere length can be determined to be ~2 μm. Scale bar: 10 μm. **k, l** Reconstructed phase images of the regions of **g** and **j**, respectively. With space gating, random-phase noise is suppressed, and therefore, the complex winding structures of the muscle fibers can be identified from the phase discontinuity.

way. It is used for gating out the multiply scattered wave in ideal diffraction-limited imaging based on the confocal detection or coherent aperture synthesis. Similar to deep tissue photoacoustic approaches[45,46], the conventional acousto-optic approaches[24–28] rely on both ballistic and multiply scattered waves as a whole. Therefore, the imaging resolution is set by the acoustic diffraction limit, which was ~30 μm in our experiments. However, it should be noted that their imaging depth can be larger than the proposed method because they are not subject to the problem of competition between the ballistic and multiply scattered waves. There have been a few ingenious wavefront shaping methods that can improve the spatial resolution of acousto-optic or photoacoustic approaches to the optical speckle scale, using iterative optimization[47–49] and variance-encoding[32,33]. However, these methods are easily compromised in practical situations, where the size of the speckle grain is as small as the optical wavelength or the acoustic focal profile does not have a well-defined peak. Those concepts have only been demonstrated for geometries, in which the gap between the scattering layer and the object plane is large

enough for the speckle grains to be at least one order of magnitude larger than the wavelength[32,33,47–49]. In contrast, our method, which relies on a ballistic wave for image formation, allows us to obtain the ideal optical diffraction-limited resolution for objects completely embedded within a scattering medium, where the speckle grains are fully developed and on average close to half the wavelength in size. Furthermore, our method is much less sensitive to speckle decorrelation than acousto-optic wavefront manipulation techniques because the dynamic motion of the scatters affects the ballistic wave much less than the multiply scattered wave.

Because of the two-dimensional nature of space gating, the noise suppression factor $\eta$ can be quadratically improved by reducing the size of the gating window $w_{SG}$. Therefore, the use of higher frequency acoustic waves or second-harmonic acousto-optic interactions would greatly improve the imaging depth, although the reduced acousto-optic modulation efficiency may potentially hinder the proper measurement of acousto-optically modulated optical wave. The imaging depth can also be greatly

improved by choosing a probe beam of longer wavelength at which $l_s$ is larger. First, it allows us to detect the ballistic wave at a proportionally larger $L$ because the intensity of the ballistic wave follows the Beer–Lambert law dictated by $L/l_s$. Secondly, and more interestingly, the effect of space gating would quadratically increase with $L$ due to the associated increase in the spatial extent of the multiply scattered wave. Although our proof-of-concept experiments were performed at the 532 nm wavelength, where $l_s$ is relatively small for biological tissues[50], the use of a longer-wavelength source would substantially increase the absolute imaging depth for biological applications. This space gating technique could also be adopted for other epi-detection configurations for more diverse applications in biological studies.

The resolution of the demonstrated imaging method is set by the diffraction limit of optical system. In the present study, the diffraction-limited resolution of 1.5 μm was set by geometric restrictions of the focused laser beam and acoustic transducer. The use of a physically smaller acoustic transducer would allow a higher numerical aperture (NA) for optical imaging. Novel aberration correction methods reported in previous studies could also be incorporated to retain submicron imaging resolution even for aberrating biological specimens[14–17]. In our experiments, the imaging speed is limited to 10 Hz per point by the laser repetition rate, the line scan time of the rolling shutter of the camera, and the scheme for the holographic measurement, but it can be improved up to 1000 Hz per point without much technical hurdles. Because the acoustic propagation time from the transducer to the acoustic spot is ~4 μs, the laser repetition rate can be increased up to 250 kHz while ensuring a single space gating for each optical pulse. Therefore, the camera exposure can be reduced down <1 ms from the current value of 10 ms because typically 100 laser pulses are sufficient for the accurate complex field measurement. The imaging speed can be further increased by twofold and fourfold, respectively, using a global-shutter camera and an off-axis holography[51]. In aggregate, those efforts will lead to ~100-fold improvement in imaging speed, which would be sufficiently fast for soft-tissue imaging.

To conclude, the imaging depth of microscopy has long been set by the ability of existing gating methods to reject multiply scattered waves. It has been particularly difficult to apply the method of phase imaging to the case, where transparent biological cells are fully embedded inside a scattering medium due to its susceptibility to multiple scattering. The proposed concept of space gating is a novel and independent gating scheme, which can effectively reject the multiply scattered wave that bypasses conventional gating operations (see Supplementary Note 3 to see when space gating is particularly beneficial). By taking the full advantage of this space gating, we could realize phase imaging of biological cells and fine tissue morphologies embedded within a thick biological tissue. Given that the space gating can be combined with all the existing gating methods for the optimal rejection of multiply scattered waves, further development and use of space gating will provide an important step toward reaching the ultimate imaging depth set by the detection limit of ballistic waves. And its capability of phase imaging in a thick scattering medium will facilitate the studies of the native physiology of biological cells within deep tissues.

## Methods

**Confocal imaging setup with acousto-optic space gating.** For confocal imaging, we sampled the modulated signals at the camera pixel conjugate to the focused illumination. This confocal configuration is in effect identical to the conventional confocal scheme based on a physical pinhole. The NAs of the objective lenses on the illumination and detection paths were 0.18, setting the diffraction limit resolution of 1.5 μm. Three cycles of the focused acoustic wave whose frequency was $f_{US} = 50$ MHz was temporally synchronized with a 532-nm laser pulse of 7 ns width at a repetition rate of 40 kHz. The frequency bandwidth of the transducer was as

wide as 40 MHz, which is wide enough to generate the short-pulsed sine wave with the correlation with respect to the ideal three-cycle sine wave >90 %. The NA of acoustic transducer was 0.47. With the acoustic pressure of a few megapascals, the spatial-peak-temporal-average intensity was ~150 mW$^2$ cm$^{-2}$, which is well below the safety limit of 720 mW$^2$ cm$^{-2}$ for biological applications.

Although the interferometric confocal detection provides the phase map of the ballistic wave, the phase drift during the focal scanning deteriorates the phase image of the object. Therefore, to achieve quantitative phase imaging, we switch the illumination beam to a plane wave and then vary the incidence angle for coherent aperture synthesis, where the coherent (i.e., both amplitude and phase) image is synthesized in such a way that the ballistic wave is collectively accumulated[12,52]. Note that once the ballistic wave has been properly accumulated for every incidence angle constituting the focused beam in confocal detection, the signal to noise ratio and the imaging resolution of the coherent aperture synthesis is identical to that of the confocal method. However, in most of our experiments for amplitude objects, we used the confocal scheme shown in Fig. 2a because it provides a higher signal to noise ratio for the initial detection of the ballistic wave before reconstructing the image.

**Measurement of spatial extent of $R$ and $R_{SG}$.** We illuminated a transparent sample composed of PAA gel with a plane wave. For the measurement of $R$ without space gating, we switched off the reference beam and the focused acoustic beam, and then summed the intensity maps measured over 900 incidence angles. For the measurement of $R_{SG}$ with space gating, we performed the interferometric detection of the acousto-optically modulated waves for 900 incidence angles and summed the intensity of the measured complex fields.

**Measurement of transfer functions.** To measure the illumination transfer function $|T_i(\mathbf{r}_o; \mathbf{r}_i)|^2$, we recorded the intensity map on the object plane using the camera shown in Fig. 2 while removing the scattering sample in the detection path. To measure the detection transfer function $|T_d(\mathbf{r}_o; \mathbf{r}_d)|^2$, we used the reciprocity of light propagation and the symmetry of our optical system. Based on reciprocity, the detection transfer function $|T_d(\mathbf{r}_o; \mathbf{r}_d)|^2$ is identical to the intensity map on the object plane for a virtual source placed at the detector point $\mathbf{r}_d$. Therefore, we removed the scattering sample from the illumination path and flipped the entire sample with respect to the object plane to take advantage of the symmetry between the input and output sides of our system. Finally, similar to the measurement of $|T_i(\mathbf{r}_o; \mathbf{r}_i)|^2$, we recorded the intensity map on the object plane while illuminating the flipped sample with a focused beam.

**Calculation of modulation efficiency.** The measured interference intensity at the $k^{th}$ phase step ($k$ is an integer number $\in [0, 3]$) can be expressed as $I_k = \left| E^{ref} \exp\left(i\frac{\pi}{2}k\right) + E^{sam} \right|^2 = \left| E^{ref} \exp\left(i\frac{\pi}{2}k\right) + E^{sam}_{unmod} + E^{sam}_{mod} \right|^2$ (ref. [31]), where $E^{ref}$ and $E^{sam}$ are the complex amplitudes of the reference and sample waves, respectively, and $E^{sam}_{unmod}$ and $E^{sam}_{mod}$ are the unmodulated and modulated components of the sample wave, respectively. Then, the modulation efficiency is defined as $\left| E^{sam}_{mod} \right|^2 / \left| E^{sam} \right|^2$. Considering the camera exposure is much longer than the acoustic oscillation period, the two interference terms involving $E^{sam}_{unmod}$ are averaged out to a negligible level due to their oscillation at the acoustic frequency. Therefore, $I_k$ can be written as $\left| E^{ref} \right|^2 + \left| E^{sam} \right|^2 + 2\left| E^{ref} \right| \left| E^{sam}_{mod} \right| \cos\left(\phi + \frac{\pi}{2}k\right)$, where $\phi$ is the relative phase between $E^{ref}$ and $E^{sam}_{mod}$. Finally, the modulation efficiency is given by $\left\{ \frac{|(I_2 - I_0) + i(I_3 - I_1)|}{4} \right\}^2 / \left( \left| E^{ref} \right|^2 \left| E^{sam} \right|^2 \right)$.

**Preparation of scattering layers.** To fabricate a scattering layer, a PDMS solution was thoroughly mixed with ZnO particles at a fixed concentration. The mixture was then transferred to a Petri dish and coated uniformly on the dish using a spin coater. Finally, the PDMS was cured at 60 °C. The scattering mean free path $l_s$ of the layer was 21 μm, which was measured by the ballistic transmission through the two distant diaphragms. The layer thickness was controlled by varying the volume of the PDMS mixture transferred to the dish and measured by a conventional bench-top microscope. The thickness ranged between 150 and 290 μm.

**Imaging procedure of amplitude objects.** The focused illumination beam was scanned over $16.1 \times 16.1$ μm$^2$ with a step size of 0.54 μm using a pair of galvanometer mirrors. This resulted in 900 illumination spots. The confocal image of the object was then reconstructed from the intensity recordings at the detector pixels conjugate to the illumination point. The amplitude objects used were 2-μm gold-coated silica microspheres with transmittance of ~10 % at 532-nm wavelength.

**Calculation of ratio $\tau$ and $\tau_{SG}$ using PSFs.** The detected confocal intensity $\langle |E(\mathbf{r}_d = \mathbf{r}_i; \mathbf{r}_i)|^2 \rangle$ equals $\langle |E_S(\mathbf{r}_d = \mathbf{r}_i; \mathbf{r}_i)|^2 \rangle + \langle |E_M(\mathbf{r}_d = \mathbf{r}_i; \mathbf{r}_i)|^2 \rangle$ because the cross term between the ballistic and multiply scattered wave converges to 0, with an ensemble average denoted by $\langle \rangle$. $\langle |E_M(\mathbf{r}_d = \mathbf{r}_i; \mathbf{r}_i)|^2 \rangle$ here can be separately determined by the intensity in the vicinity ($\mathbf{r}_d \sim \mathbf{r}_i$) of the illumination spot because $\langle |E_M(\mathbf{r}_d; \mathbf{r}_i)|^2 \rangle$ varies slowly with $\mathbf{r}_d$.

The ratio of the ballistic waves to the multiply scattered waves was calculated using two methods, depending on the visibility of the focused ballistic wave. When the focused spot was clearly visible (i.e., the peak to background ratio was >5), the detected confocal intensity $\langle |E(\mathbf{r}_d = \mathbf{r}_i; \mathbf{r}_i)|^2 \rangle$ and the $\langle |E_M(\mathbf{r}_d \sim \mathbf{r}_i; \mathbf{r}_i)|^2 \rangle$ could be quantified directly from the PSFs. $\tau$ and $\tau_{SG}$ were respectively determined as $\frac{[\langle |E(\mathbf{r}_d = \mathbf{r}_i; \mathbf{r}_i)|^2 \rangle - \langle |E_M(\mathbf{r}_d \sim \mathbf{r}_i; \mathbf{r}_i)|^2 \rangle]}{\langle |E_M(\mathbf{r}_d \sim \mathbf{r}_i; \mathbf{r}_i)|^2 \rangle}$ and $\frac{[\langle |E^{SG}(\mathbf{r}_d = \mathbf{r}_i; \mathbf{r}_i)|^2 \rangle - \langle |E_M^{SG}(\mathbf{r}_d \sim \mathbf{r}_i; \mathbf{r}_i)|^2 \rangle]}{\langle |E_M^{SG}(\mathbf{r}_d \sim \mathbf{r}_i; \mathbf{r}_i)|^2 \rangle}$. However, when the focused spot was not clearly visible, such as in Fig. 2g, $\langle |E_S(\mathbf{r}_d = \mathbf{r}_i; \mathbf{r}_i)|^2 \rangle$ cannot be precisely estimated by $\langle |E(\mathbf{r}_d = \mathbf{r}_i; \mathbf{r}_i)|^2 \rangle - \langle |E_M(\mathbf{r}_d \sim \mathbf{r}_i; \mathbf{r}_i)|^2 \rangle$. In this case, $\langle |E_S(\mathbf{r}_d = \mathbf{r}_i; \mathbf{r}_i)|^2 \rangle$ was estimated by $I_0 \exp(-L_{tot}/l_s)$, where $I_0$ is the measured peak intensity through a transparent specimen. Then, $\tau$ was estimated as $\frac{I_0 \exp(-L_{tot}/l_s)}{\langle |E_M(\mathbf{r}_d \sim \mathbf{r}_i; \mathbf{r}_i)|^2 \rangle}$.

**Preparation of embedded objects**. A thin PAA gel layer mixed with gold-coated microspheres was sandwiched between two 3-mm-thick PAA gel slabs containing a 0.8% fat emulsion (Intralipid). The total optical thickness $L_{tot}/l_s$ of the 6-mm-thick PAA gel was measured to be 21.0. Similarly, we prepared human red blood cells sandwiched between PAA gels with a 0.8% fat emulsion ($L_{tot}/l_s \sim 21$) to mimic biological conditions. We recorded the speckle pattern at the object plane with a 1.4-NA objective lens in the absence of a PAA slab on the detection side and determined the average grain size at the object plane from the FWHM of the autocorrelation function of the speckle pattern.

**Reconstruction of extended field-of-view image**. To image different parts of the whole-body zebrafish, the sample cuvette containing the zebrafish was mounted on a three-axis-motorized translation stage. For each sample position, individual images were captured and processed using coherent aperture synthesis following the procedure described in the Results section. The size of the individual image along x- and y-axes was 130 μm × 130 μm without space gating and 50 μm × 80 μm with space gating, respectively, set by the number of active sensor elements and the size of space gating. To acquire a large field-of-view image, the zebrafish was translated with a step size of 65 μm × 65 μm and 25 μm × 40 μm along x- and y-axes, respectively, without and with space gating. Finally, the individual images were coherently combined into a complex, large field-of-view image based on the autocorrelation of the overlapped area between the adjacent images.

**Reporting summary**. Further information on research design is available in the Nature Research Reporting Summary linked to this article.

## Code availability

The MATLAB codes used in this work are available from the corresponding authors upon request.

## Data availability

The datasets acquired for this study are available from the corresponding authors upon request.

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

## Acknowledgements

This research was supported by IBS-R023-D1. W.K.L. and J.-S.L. were supported by National Research Foundation of Korea (NRF-2016R1A5A1010148). M.J. was supported by TJ Park Science Fellowship of POSCO TJ Park Foundation. The fixed zebrafish specimen was provided from Prof. Hae-Chul Park at Korea University Ansan Hospital.

## Author contributions

M.J. and W.C. conceived the initial idea. M.J. developed the theoretical modeling, designed the experiments, and analyzed the experimental data with the help of H.K. and W.C. H.K. prepared the sample and carried out the experiments with the help of M.J., H.K., and J.H.H. prepared the zebrafish specimen. W.K.L. and J.-S.L. fabricated the gold-coated silica microspheres. All authors contributed to writing the manuscript. W.C. supervised the project.

## Competing interests

The authors declare no competing interests.
