## [Peer Review File · Nature Communications]

Reviewers' comments:

Reviewer #1 (Remarks to the Author):

The authors have done a good job of addressing all of the reviewers extensive comments. The revised manuscript is considerably improved over the original submission. I support publication of this important advance in deep tissue imaging.

Reviewer #2 (Remarks to the Author):

First all, I would like to thank the authors for carefully and thoroughly addressing the technical concerns raised by me and other reviewers. The revised manuscript is much improved in clarity and thoroughness. I also applaud the authors' efforts to explain the key technical innovations of this method compared with other acoustic-optical methods that use ultrasound signals to modulate the photons and improve the contrast to noise of the image in deeper tissue, in this case, the ratio between the ballistic photons and multi-scattered photons for gated imaging. It is now clear to me that the combination of the spatial gating provided by the focused ultrasound modulation with the traditional gating mechanisms. The experiments on phantoms and zebrafish have demonstrated the feasibility of this method and their potential use in biomedical research. It is indeed very useful to see how the acoustic modulation is performed to spatially 'catch' the optical focus. I think the revised manuscript has addressed most of my concerns and may be accepted for publication in Nature Communication, with a few minor concerns listed below.

1. While the authors have correctly pointed out that traditional acoustic-optical modulation only provides ultrasound determined resolution, I am not sure if the authors are aware of the work on wavefront shaping that has achieved optical diffraction limit with acoustic modulation, such as the work by the Yang group at Caltech, <https://www.nature.com/articles/nphoton.2013.31>. It will be interesting to discuss the pro and cons of the two methods.
2. The authors are fairly clear on the general imaging principles. More detailed discussion or estimation on the theoretical penetration limit using this spatial gating will be useful, given the realistic optical and acoustic conditions of biological tissue, such as mouse brain.
3. What is the limiting factor for using higher frequency ultrasound transducers? Since only 1 mm or so penetration is needed, high frequency ultrasound at the level of hundred MHz should be available. or does it actually help at all to have a smaller acoustic focus?

Reviewer #3 (Remarks to the Author):

The manuscript has been improved a lot since I refereed it last time. All of my previous criticisms have been addressed, and now the manuscript is clear and legible, with a well defined scope and story. In particular the "Principles" section is now much better.

I am happy to recommend publication, and I am sure that this work will attract the attention of people doing gated imaging in biological media. I only have a couple of small comments:

* The "space gating" described by the authors is technically not new (as the author themselves acknowledge in lines 192-194 of the revised manuscript). What is new is the realization that it can be adapted in such a way that it complements and improves other gating techniques. This is enough, and there is no need to claim more than that to have a very interesting paper.

* The equations on line 138 and 139-140 definitively deserve to be numbered equations on their own line, instead of being cramped in-line.

Reviewer #1:

The authors have done a good job of addressing all of the reviewers extensive comments. The revised manuscript is considerably improved over the original submission. I support publication of this important advance in deep tissue imaging.

We thank the reviewer for acknowledging the improvement of our manuscript and supporting its publication to Nature Communications.

Reviewer #2:

First all, I would like to thank the authors for carefully and thoroughly addressing the technical concerns raised by me and other reviewers. The revised manuscript is much improved in clarity and thoroughness. I also applaud the authors' efforts to explain the key technical innovations of this method compared with other acoustic-optical methods that use ultrasound signals to modulate the photons and improve the contrast to noise of the image in deeper tissue, in this case, the ratio between the ballistic photons and multi-scattered photons for gated imaging. It is now clear to me that the combination of the spatial gating provided by the focused ultrasound modulation with the traditional gating mechanisms. The experiments on phantoms and zebrafish have demonstrated the feasibility of this method and their potential use in biomedical research. It is indeed very useful to see how the acoustic modulation is performed to spatially 'catch' the optical focus. I think the revised manuscript has addressed most of my concerns and may be accepted for publication in Nature Communication, with a few minor concerns listed below.

We deeply appreciate the reviewer's acknowledging our effort to clarify the key innovation and favorable recommendation. In the following, we addressed the additional concerns raised by the reviewer.

R2C1. While the authors have correctly pointed out that traditional acoustic-optical modulation only provides ultrasound determined resolution, I am not sure if the authors are aware of the work on wavefront shaping that has achieved optical diffraction limit with acoustic modulation, such as the work by the Yang group at Caltech, <https://www.nature.com/articles/nphoton.2013.31>. It will be interesting to discuss the pro and cons of the two methods.

Thank you for bringing up this important point. As the reviewer noted, a number of novel approaches have been proposed in the field of wavefront shaping to overcome the traditional resolution limit (i.e. the acoustic diffraction limit) of acousto-optic imaging methods. We are fully aware of those recent efforts, including the variance-encoding method (Ref. 37 of our manuscript) the reviewer referred to. The fundamental difference between the 'space' gating approach and the variance-encoding method is that the former relies on the ballistic waves and the latter primarily relies on the multiply scattered waves. Because of this difference, our method robustly provides an ideal optical diffraction-limited resolution while the penetration depth is

relatively shallow compared to the variance-encoding method. On the contrary, the variance-encoding method and other wavefront shaping methods aiming at overcoming the acoustic diffraction limit can hardly achieve the resolution of $\sim 1 \mu\text{m}$. According to the recent finding (Ref. 38 of our manuscript), a large number of measurements are required to guarantee statistical robustness of variance-encoding methods, and this number grows cubically with the number of optical speckle grains within the ultrasound focus. Considering the number of optical modes of $\sim 10^3$ - 10^4 within the ultrasound focus, our methods provides an ideal optical diffraction-limited imaging with $\sim 10^3$ - 10^4 measurements, while the variance-encoding methods requires $\sim 10^9$ - 10^{12} measurements. In fact, we already discussed the pros and cons of our approach compared to the variance-encoding method in the Discussion section of the original manuscript. We reiterate the corresponding paragraph below to assist reviewer's verification.

“The proposed space gating method is the first acousto-optic imaging approach relying on the selective and coherent detection of the ballistic waves. Therefore, its resolution is dictated by the ideal diffraction limit of the optical system, rather than the diffraction limit of the acoustic system. Although our scheme of space gating shares some components with the conventional ultrasound-modulated optical tomography²⁶⁻³⁰, our space-gated microscopy uses the acousto-optic effect in a completely different way. It is used for gating out the multiply scattered wave in ideal diffraction-limited imaging based on the confocal detection or coherent aperture synthesis. Similar to deep-tissue photoacoustic approaches^{51,52}, the conventional acousto-optic approaches²⁶⁻³⁰ rely on both ballistic and multiply-scattered waves as a whole. Therefore, the imaging resolution is set by the acoustic diffraction limit, which was around $30 \mu\text{m}$ in our experiments. However, it should be noted that their imaging depth can be larger than the proposed method because they are not subject to the problem of competition between the ballistic and multiply scattered waves. There have been a few ingenious wavefront shaping methods that can improve the spatial resolution of acousto-optic or photoacoustic approaches to the optical speckle scale using iterative optimization⁵³⁻⁵⁵ and variance-encoding^{37,38}. However, these methods are easily compromised in practical situations where the size of the speckle grain is as small as the optical wavelength or the acoustic focal profile does not have a well-defined peak. Those concepts have only been demonstrated for geometries in which the gap between the scattering layer and the object plane is sufficiently large that the speckle grains are at least one order of magnitude larger than the wavelength^{37,38,53-55}. In contrast, our method, which relies on a ballistic wave for image formation, allows us to obtain the ideal optical diffraction-limited resolution for objects completely embedded within a scattering medium, where the speckle grains are fully developed and on average close to half the wavelength in size. Furthermore, our method is much less sensitive to speckle decorrelation than acousto-optic wavefront manipulation techniques because the dynamic motion of the scatters affects the ballistic wave much less than the multiply scattered wave.”

R2C2. The authors are fairly clear on the general imaging principles. More detailed discussion or estimation on the theoretical penetration limit using this spatial gating will be useful, given the realistic optical and acoustic conditions of biological tissue, such as mouse brain.

We agree that the theoretical penetration limit of space-gated imaging in realistic conditions would be useful. The penetration limit is set by the depth where the signal to noise ratio τ_{SG} (i.e. the intensity ratio of ballistic wave to multiply scattered wave with space gating) is sufficiently larger than 1. τ_{SG} is given as the multiplication of the signal to noise ratio τ without space gating (i.e. of a conventional confocal microscopy) and the noise suppression factor η (i.e. $\tau_{SG} = \tau \times \eta$). The quantitative behavior of τ , τ_{SG} , and η depends highly on the optical properties of the sample and the configurations of the sample and imaging system. Therefore, the estimation of the absolute values of imaging depth requires case study. For instance, one example study could be found in Ref. 19 of our manuscript.

Instead of providing a full quantitative analysis on τ , τ_{SG} , and η , we mainly focused on the estimation of the effect of space gating, η , and the gain in the penetration depth relative to that of a conventional confocal microscopy. The following paragraphs in the main text made this intention clear.

“For biological tissues, Δw_{M_i} and Δw_{M_d} typically range from hundreds of microns to millimeters when $L/l_s \sim 10$. Therefore, we can expect $\eta > 100$ if the size of the space gating Δw_{SG} is as small as tens of microns, as is the case with a high-frequency acoustic focus.”

“The space gating improves the imaging fidelity by a factor of η , i.e. $\tau_{SG} = \eta \times \tau$. Considering the exponential decay of the intensity of ballistic wave, the imaging depth increases logarithmically with η . More specifically, the noise suppression effect can compensate the additional decay of ballistic wave by the increased imaging depth, i.e. $\eta \times e^{-\Delta L/l_s} = 1$, where ΔL is the gain in the imaging depth by the space gating. Therefore, η is translated into $\Delta L = l_s \times \log \eta$. For $\eta > 100$, we can expect the gain in imaging depth ΔL of more than $5l_s$.”

To provide a more realistic estimation, the penetration depth limit of a conventional confocal microscopy is $\sim 500 \mu\text{m}$ and the scattering mean free path l_s is $\sim 50 \mu\text{m}$ within a brain tissue. In such case, the space-gated imaging would increase the penetration depth by $\sim 5l_s$ ($= 250 \mu\text{m}$), resulting in the penetration depth limit of $\sim 750 \mu\text{m}$. Again, such estimation is exemplary.

R2C3. What is the limiting factor for using higher frequency ultrasound transducers? Since only 1 mm or so penetration is needed, high frequency ultrasound at the level of hundred MHz should be available. or does it actually help at all to have a smaller acoustic focus?

The limiting factor for using higher frequency ultrasound source is the attenuation of ultrasonic wave in biological tissue. The attenuation is approximately given as $0.06 [\text{dB}/\text{MHz}/\text{mm}] \times f \times d$ where f is the ultrasonic frequency in MHz and d is the depth in mm. In the ultrasound

frequency at the level of hundred MHz, the attenuation would be at the level of 10 dB, which significantly reduce the acousto-optic modulation efficiency. Considering the modulation efficiency is already as low as a few percent at the ultrasonic frequency of 50 MHz, this could potentially hinder the accurate measurement of the modulated optical wave buried in the unmodulated optical wave. In the revised manuscript, we clarified this potential challenge in increasing the ultrasonic frequency in the underlined part of Discussion section.

“Therefore, the use of higher frequency acoustic waves or second-harmonic acousto-optic interactions would greatly improve the imaging depth, although the reduced acousto-optic modulation efficiency may potentially hinder the proper measurement of acousto-optically modulated optical wave.”

As provided in Eq. (4), the smaller acoustic focus would quadratically improve the noise suppression factor η :

$$\eta = |E_M(r_d; r_i)|^2 / |E_M^{SG}(r_d; r_i)|^2 \sim \min(\Delta w_{M_i}, \Delta w_{M_d})^2 / \Delta w_{SG}^2. \#(4)$$

Therefore, if the size of space gating is reduced by β times, the noise suppression factor is increased by β^2 and the penetration depth is increased by $2\log(\beta) \times l_s$. For instance, if the noise suppression factor η is 100 with the 50-MHz ultrasound source, η will be increased to 400 with the 100-MHz ultrasound source. This improvement is translated into the penetration depth gain of $\sim 1.4 l_s$. We concisely articulated this behavior of space gating in the original Supplementary Note 2 as follows.

“ Δw_{M_i} and Δw_{M_d} are respectively the widths of the multiply scattered waves in the illumination and detection transfer functions, and Δw_{SG} is the width of the space gating set by the size of the acoustic focus. Therefore, for the imaging depth increase of more than $\alpha \times l_s$, Δw_{SG} needs to be smaller than $e^{-\alpha/2} \times \min(\Delta w_{M_i}, \Delta w_{M_d})$.”

Reviewer #3:

The manuscript has been improved a lot since I refereed it last time. All of my previous criticisms have been addressed, and now the manuscript is clear and legible, with a well defined scope and story. In particular the "Principles" section is now much better. I am happy to recommend publication, and I am sure that this work will attract the attention of people doing gated imaging in biological media. I only have a couple of small comments:

We thank the reviewer for acknowledging the improvement of our manuscript and recommending the publication. The reviewer's comments played an important role in improving the clarity of the manuscript.

R3C1. The "space gating" described by the authors is technically not new (as the author

themselves acknowledge in lines 192-194 of the revised manuscript). What is new is the realization that it can be adapted in such a way that it complements and improves other gating techniques. This is enough, and there is no need to claim more than that to have a very interesting paper.

As the reviewer pointed out, we clearly specified that the interferometric detection of ultrasonically modulated optical wave is not technically new by itself by including the statement in the lines 192-194 of the original manuscript:

“The scheme that implements the space gating is based on an interferometric detection method similar to the previously demonstrated ultrasound-modulated optical tomography”

The main innovation of our work is to introduce the novel concept of space ‘gating,’ where the acousto-optic interaction’ is used for rejecting multiply scattered waves and extending the imaging depth of ideal optical diffraction-limited imaging. In the revised manuscript, we further clarified this point in the Introduction section. We added the underlined phrase in the following.

“Here we propose a new gating scheme, called ‘space’ gating, based on the interferometric detection scheme of previous acousto-optic imaging techniques²²⁻²⁸.”

R3C2. The equations on line 138 and 139-140 definitively deserve to be numbered equations on their own line, instead of being cramped in-line.

We thank the reviewer for the valuable suggestion. We revised the text accordingly.